# Observation of spin-orbit effects with spin rotation symmetry

Alisha M. Humphries [1], Tao Wang[2], Eric R.J. Edwards[3], Shane R. Allen[1], Justin M. Shaw[3], Hans T. Nembach[3], John Q. Xiao[2], T.J. Silva[3] & Xin Fan[1]

The spin–orbit interaction enables interconversion between a charge current and a spin current. It is usually believed that in a nonmagnetic metal (NM) or at a NM/ferromagnetic metal (FM) bilayer interface, the symmetry of spin–orbit effects requires that the spin current, charge current, and spin orientation are all orthogonal to each other. Here we demonstrate the presence of spin–orbit effects near the NM/FM interface that exhibit a very different symmetry, hereafter referred to as spin-rotation symmetry, from the conventional spin Hall effect while the spin polarization is rotating about the magnetization. These results imply that a perpendicularly polarized spin current can be generated with an in-plane charge current simply by use of a FM/NM bilayer with magnetization collinear to the charge current. The ability to generate a spin current with arbitrary polarization using typical magnetic materials will benefit the development of magnetic memories.

[1] Department of Physics and Astronomy, University of Denver, Denver, CO 80210, USA. [2] Department of Physics and Astronomy, University of Delaware, Newark, DE 19716, USA. [3] Quantum Electromagnetics Division, National Institute of Standards and Technology, Boulder, CO 80305, USA. Alisha M. Humphries and Tao Wang contributed equally to this work. Correspondence and requests for materials should be addressed to X.F. (email: xin.fan@du.edu)

The interconversion between a charge current and a spin current driven by the spin–orbit interaction has been actively studied[1–10]. It has been shown that an in-plane charge current in a ferromagnet/nonmagnetic metal (FM/NM) bilayer can generate spin–orbit torque (SOT) via the bulk spin Hall effect in the NM[7] and/or from the interfacial SOEs at the FM/NM interface[11–13]. These effects can be used for magnetization switching with an in-plane charge current, with potential benefits for the development of magnetic random access memories (MRAM)[14]. The symmetry of spin-current generation by the spin Hall effect is captured by the essential phenomenology,

$$\mathbf{Q}_{\hat{\sigma}} = \frac{\hbar}{2e}\theta\big(\mathbf{j}_e \times \hat{\sigma}\big),\qquad(1)$$

where $\mathbf{j}_e$ is the in-plane charge current density, $\mathbf{Q}_{\hat{\sigma}}$ is the out-of-plane spin-current density with $\hat{\sigma}$ denoting its spin polarization, $\theta$ is the spin/charge conversion efficiency, $\hbar$ is the reduced Planck's constant, and $e$ is the electron charge. According to Eq. (1), an in-plane charge current can generate an out-of-plane flowing spin current, but only with spins polarized in-plane and perpendicular to the charge current. As such, the direct switching of a perpendicular magnetized film via the combination of the spin Hall effect and spin torque transfer (i.e., anti-damping) is not possible. To cause such switching, additional sources of broken symmetry are required, such as an intrinsic gradient of the magnetic anisotropy[15], tilting of the magnetization by an external magnetic field relative to the interface normal[16, 17], or an effective exchange field[18, 19]. Even then, if tilting of the magnetization facilitates the switching process, the SOT must necessarily overcome both the torque due to anisotropy as well as that of the damping. As such, the efficiency of such a switching process is necessarily compromised.

A spin current with an out-of-plane polarization can switch a perpendicular magnetization via the anti-damping process without the need to tilt the magnetization, but for this spin–orbit effect, additional symmetry breaking is required for this to happen. For example, MacNeill et al.[20] recently showed that a spin current with unconventional symmetry can be generated in a WTe$_2$/Permalloy bilayer due to the unique crystal symmetry of the transition-metal dichalcogenide. Alternatively, Taniguchi et al.[21] have proposed that an out-of-plane polarized spin current can be generated via the combination of the anomalous Hall effect in a FM with tilted magnetization and the spin-filtering effect. More generally, Amin and Stiles have predicted that spin–orbit scattering of an in-plane charge current at a FM/NM interface can give rise to a spin current with an arbitrary spin polarization, because of the interaction between spins and the magnetic order at the interface[22]. One possible microscopic mechanism consistent with such a prediction is the case where

spin polarization of a spin current generated near the FM/NM interface precesses about the magnetization. Although transverse spins rapidly dephase in a FM[23, 24], this is not necessarily the case at the FM/NM interface or when FM is very thin. Therefore, from a purely phenomenological point of view, we might expect a source of spin current described by

$$\mathbf{Q}_{\hat{\sigma}}^{R} = \frac{\hbar}{2e}\theta^{R}\mathbf{j}_e \times (\hat{\mathbf{m}} \times \hat{\sigma}),\qquad(2)$$

where $\theta^R$ is the spin/charge conversion efficiency for the SOE with the rotated spin symmetry. In this sense, the generation of a spin current described by Eq. (2) is loosely analogous to the rotation of the polarization of light by the Faraday effect.

As shown in Fig. 1a, when an in-plane charge current passes through a FM/NM interface, an out-of-plane propagating spin current can be generated with two components in accordance with both Eqs. (1) and (2). It should be emphasized that Eq. (2) describes an effect that is inherently different from the spin filtering proposed by Taniguchi et al.[21] The polarization of the spin current generated via spin filtering is always polarized collinear with the magnetization, whereas the spin current due to spin rotation is always polarized orthogonal to the magnetization.

Similarly, as illustrated in Fig. 1b, a spin current $\mathbf{Q}_{\hat{\sigma}}$ that flows out-of-plane in a FM/NM bilayer can generate two in-plane charge currents via the spin galvanic effect (SGE), one with $\mathbf{j}_e$ in the direction $\mathbf{Q}_{\hat{\sigma}} \times \hat{\sigma}$, and one with $\mathbf{j}_e^R$ in the rotated direction $-\mathbf{Q}_{\hat{\sigma}} \times (\hat{\mathbf{m}} \times \hat{\sigma})$. This process can be mathematically described as

$$\begin{aligned}\mathbf{j}_e &= \frac{2e}{\hbar}\theta\,\mathbf{Q}_{\hat{\sigma}} \times \hat{\sigma},\\ \mathbf{j}_e^R &= -\frac{2e}{\hbar}\theta^R\mathbf{Q}_{\hat{\sigma}} \times (\hat{\mathbf{m}} \times \hat{\sigma}),\end{aligned}\qquad(3)$$

where the negative sign in the second equation is necessary to satisfy the Onsager relation as discussed in Supplementary Note 1.

Here, we present observations of spin–orbit effects with spin-rotation symmetry as described by Eqs. (2) and (3). In the following section we show experimental results that corroborate the SOE with spin-rotation symmetry by use of current-induced SOT and spin Seebeck effect (SSE)-driven SGE measurements.

## Results

**Current-induced spin–orbit torque measurements.** First, we present the detection of the spin current with rotated spins generated near an interface between Cu and a perpendicular magnetized layer (PML), as described by Eq. (2). The test sample is a multilayer with the structure seed/PML/Cu (3)/Py(2)/Pt(3), and the control sample has the structure seed/PML/Cu(3)/TaO$_x$(3)/Py(2)/Pt(3), where seed = Ta(2)/Cu(3), PML = [Co$_{90}$Fe$_{10}$(0.16)/Ni(0.6)]$_8$/Co$_{90}$Fe$_{10}$(0.16), Py = Ni$_{80}$Fe$_{20}$, and the numbers in parentheses are nominal thicknesses in nanometers. The Py layer is the spin-current detector. The TaO$_x$ insulating layer in the control sample blocks the flow of spin current between the PML and Py layers. The electrical and magnetic properties of the test sample are shown in the Supplementary Note 2.

The measurement geometry is shown in Fig. 2a. The sample is patterned into a 50 μm × 50 μm strip and connected by large gold contact pads. The sample is then placed onto a motion stage, which allows easy focusing and laser scanning over the sample. A laser is incident perpendicularly onto the sample through a ×20 objective. In this measurement, an external field is swept along the x-direction. We manually rotate the sample together with the applied current in the film plane. The out-of-plane and in-plane magnetization rotation in Py due to current-induced torques are

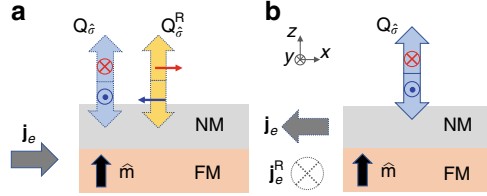

**Fig. 1** Spin–orbit effects with conventional and rotated symmetries. **a** Spin currents generated from a charge current near the FM/NM interface. The red and dark blue arrows represent spins. The light blue and yellow arrows represent the spin current with conventional symmetry, $\mathbf{Q}_{\hat{\sigma}}$, and the spin current with spin-rotation symmetry, $\mathbf{Q}_{\hat{\sigma}}^R$, respectively. The gray arrow represents the charge current $\mathbf{j}_e$. **b** A sketch of the reciprocal process to illustrate how the conventional and rotated charge currents are generated by a pure spin current near the FM/NM interface

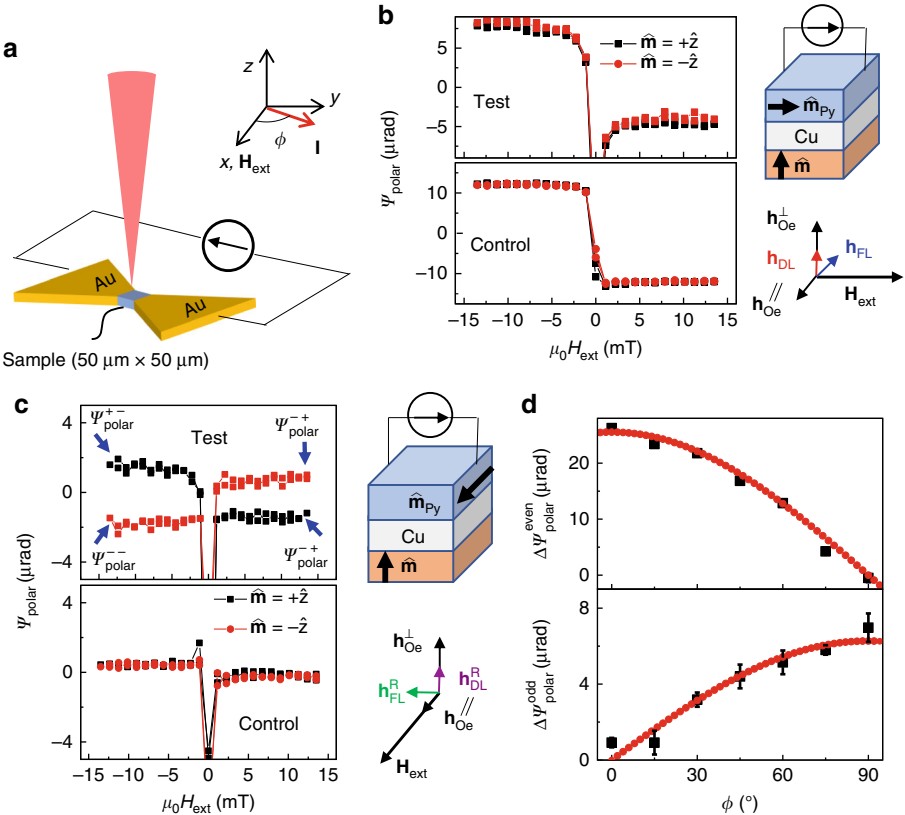

**Fig. 2** Current-induced damping-like torque. **a** Experimental configuration for measuring the SOTs by MOKE magnetometry. **b** The polar MOKE response, $\psi_{\text{polar}}$, measured in the test and control samples when current is applied parallel to $H_{\text{ext}}$ ($\phi = 0°$). No dependence on $\hat{\mathbf{m}}$ is observed. The measurement geometry (for simplicity we omit the Pt capping and seed layers) and the effective fields applied on the sample are also shown. **c** The polar MOKE response measured in the test and control samples when current is applied perpendicular to $H_{\text{ext}}$ ($\phi = 90°$). In the test sample, the polar MOKE response is reversed when $\hat{\mathbf{m}}$ is reversed. In contrast, the polar MOKE response in the control sample has little dependence on $\hat{\mathbf{m}}$. The weak hysteresis-like signal in the control sample is likely due to small misalignment ($\sim 1.5°$) of $H_{\text{ext}}$. **d** The angle dependence of $\Delta\psi_{\text{polar}}^{\text{even}}$ and $\Delta\psi_{\text{polar}}^{\text{odd}}$ of the test sample, where we define $\Delta\psi_{\text{polar}}^{\text{even}} = \left(\psi_{\text{polar}}^{++} - \psi_{\text{polar}}^{+-}\right) + \left(\psi_{\text{polar}}^{-+} - \psi_{\text{polar}}^{--}\right)$ and $\Delta\psi_{\text{polar}}^{\text{odd}} = \left(\psi_{\text{polar}}^{++} - \psi_{\text{polar}}^{+-}\right) - \left(\psi_{\text{polar}}^{-+} - \psi_{\text{polar}}^{--}\right)$. The first superscript in $\psi_{\text{polar}}^{++}$ denotes the sign of $\hat{\mathbf{m}}$ and the second superscript denotes the sign of $\hat{\mathbf{m}}_{\text{Py}}$. The red lines are fittings using $\Delta\psi_{\text{polar}}^{\text{even}} = 25.5\cos\left(\frac{\pi}{180}\phi\right)$ and $\Delta\psi_{\text{polar}}^{\text{odd}} = 6.3\sin\left(\frac{\pi}{180}\phi\right)$ and the error bars represent the standard deviation calculated from the linear fitting in **b** used to calculate $\Delta\psi_{\text{polar}}^{\text{even}}$ and $\Delta\psi_{\text{polar}}^{\text{odd}}$

measured by the polar magneto-optic-Kerr-effect (MOKE) and quadratic MOKE magnetometry, in which the incident light has a linear polarization of 45° and 0° from the $x$-direction, respectively.

According to Eq. (2), an in-plane charge current $\mathbf{j}_e$ generates spin currents with three components that exert torques on the Py magnetization, $\hat{\mathbf{m}}_{\text{Py}}$: $\mathbf{Q}_{\hat{\sigma}}$ with $\hat{\sigma} \| (\hat{\mathbf{j}}_e \times \hat{\mathbf{z}})$ due to the SOE with conventional symmetry near the Pt/Py and PML/Cu interfaces, $\mathbf{Q}_{\hat{\sigma}}^{\text{R}}$ with $\hat{\sigma} \| \hat{\mathbf{m}} \times (\hat{\mathbf{j}}_e \times \hat{\mathbf{z}})$. due to the SOE with spin-rotation symmetry near the PML/Cu interface, and $\mathbf{q}_{\hat{\sigma}}^{\text{R}}$ with $\hat{\sigma} \| \hat{\mathbf{m}}_{\text{Py}} \times (\hat{\mathbf{j}}_e \times \hat{\mathbf{z}})$. due to the SOE with spin-rotation symmetry near the Cu/Py and Py/Pt interfaces. Here, $\hat{\mathbf{m}}$ is the unit magnetization vector of the PML and $\hat{\mathbf{j}}_e$ is the unit vector along the direction of the applied current. In general, a spin current with spin polarization $\hat{\sigma}$ can exert two types of spin torques on the Py magnetization: a damping-like (DL) torque in the direction of $\left(\hat{\mathbf{m}}_{\text{Py}} \times \hat{\sigma}\right) \times \hat{\mathbf{m}}_{\text{Py}}$, which is equivalent to an effective field in the direction of $\hat{\mathbf{m}}_{\text{Py}} \times \hat{\sigma}$, and a field-like (FL) torque in the direction of $\hat{\sigma} \times \hat{\mathbf{m}}_{\text{Py}}$, which is equivalent to an effective field in the direction of $\hat{\sigma}$. Therefore, there are four effective fields due to the various spin–orbit effects that act on the Py magnetization: (1) $h_{\text{DL}}\hat{\mathbf{m}}_{\text{Py}} \times (\hat{\mathbf{j}}_e \times \hat{\mathbf{z}})$ due to the DL torque from $\mathbf{Q}_{\hat{\sigma}}$ and the FL torque from $\mathbf{q}_{\hat{\sigma}}^{\text{R}}$; (2) $h_{\text{FL}}(\hat{\mathbf{j}}_e \times \hat{\mathbf{z}})$ due the FL torque from $\mathbf{Q}_{\hat{\sigma}}$ and the

DL torque from $\mathbf{q}_{\hat{\sigma}}^{\text{R}}$; and (3) $h_{\text{DL}}^{\text{R}}\hat{\mathbf{m}}_{\text{Py}} \times \left[\hat{\mathbf{m}} \times (\hat{\mathbf{j}}_e \times \hat{\mathbf{z}})\right]$ and $h_{\text{FL}}^{\text{R}}\hat{\mathbf{m}} \times (\hat{\mathbf{j}}_e \times \hat{\mathbf{z}})$ due to the DL and field-torques from $\mathbf{Q}_{\hat{\sigma}}^{\text{R}}$, respectively. It should be emphasized the possible effective fields due to the Rashba–Edelstein spin–orbit effects at the Cu/Py and Pt/Py interfaces will share the same symmetry as $h_{\text{DL}}\hat{\mathbf{m}}_{\text{Py}} \times (\hat{\mathbf{j}}_e \times \hat{\mathbf{z}})$ and $h_{\text{FL}}(\hat{\mathbf{j}}_e \times \hat{\mathbf{z}})$, thus not discussed separately.

Besides the SOTs, an in-plane current also generates a uniform in-plane Oersted field $h_{\text{Oe}}^{\|}$, and a spatially varying out-of-plane Oersted field $h_{\text{Oe}}^{\perp}$. In the limit where the current-induced SOT is small relative to the torque due to the applied field and the demagnetizing field, the out-of-plane and in-plane components of the Py magnetization reorientation are, respectively, given by

$$m_{\text{Py}}^{\perp} = \frac{h_{\text{DL}}\left[\hat{\mathbf{m}}_{\text{Py}} \times (\hat{\mathbf{j}}_e \times \hat{\mathbf{z}})\right] \cdot \hat{\mathbf{z}} + h_{\text{DL}}^{\text{R}}\left[\hat{\mathbf{m}}_{\text{Py}} \times (\hat{\mathbf{m}} \times (\hat{\mathbf{j}}_e \times \hat{\mathbf{z}}))\right] \cdot \hat{\mathbf{z}} + h_{\text{Oe}}^{\perp}}{|H_{\text{ext}}| + M_{\text{eff}}}$$

(4)

and

$$m_{\text{Py}}^{\|} = \frac{h_{\text{FL}}\left(\hat{\mathbf{x}} \cdot \hat{\mathbf{j}}_e\right) + h_{\text{FL}}^{\text{R}}\left[\hat{\mathbf{x}} \cdot (\hat{\mathbf{m}} \times \hat{\mathbf{j}}_e)\right] + h_{\text{Oe}}^{\|}\left(\hat{\mathbf{x}} \cdot \hat{\mathbf{j}}_e\right)}{|H_{\text{ext}}|}$$

(5)

where $H_{\text{ext}}$ is the applied external magnetic field, and $M_{\text{eff}}$ is the effective Py demagnetizing field along the $z$-direction. A more

thorough analysis of spin–orbit torques on Py is shown in the Supplementary Note 3, which takes into consideration of the spin-filtering effect due to the magnetization tilting of PML.

We detect $m_{Py}^{\perp}$ by use of polar MOKE magnetometry[25], which results in the polarization rotation of $\psi_{polar}$ of linearly polarized incident light. The three terms in Eq. (4) can be distinguished by their dependence on $\hat{m}_{Py}$ and $\hat{m}$. In the measurement geometry with $j_e \| \hat{m}_{Py}$, the second term in Eq. (4) is zero. As shown in Fig. 2b, signals of $\psi_{polar}$ in both the test and control samples resemble the Py magnetization hysteresis, which can be understood from the first term in Eq. (4). The signal is independent of $\hat{m}$. In the third term in Eq. (4), the Oersted field, $h_{Oe}^{\perp}$, is independent of $\hat{m}$ and $\hat{m}_{Py}$ and spatially varies transverse to $j_e$. We use $h_{Oe}^{\perp}$, calculated by use of the Biot–Savart law, to calibrate the magnitude of $h_{DL}$ and $h_{DL}^R$, both of which depend on $\hat{m}_{Py}$ and are uniform across the sample. We scan the laser across the sample in the direction perpendicular to the applied current and isolate the polar MOKE response that is independent to $\hat{m}_{Py}$. This component of the polar MOKE signal is proportional to $h_{Oe}^{\perp}$ and its spatial distribution is fitted with the Biot–Savart law[26]. This fitting is used to establish the sensitivity of the polar MOKE measurements, i.e., we determine the ratio of the polar MOKE signal and the corresponding out-of-plane field, which can then be used to evaluate $h_{DL}$ and $h_{DL}^R$ in subsequent measurements. From the calibration, we estimate that $h_{DL}$ is about $120 \pm 12$ $Am^{-1}$ in the test sample, when the current density through Pt is about $1.2 \times 10^{10}$ $Am^{-2}$ and that through the PML is about $3.8 \times 10^{10}$ $Am^{-2}$.

In the measurement geometry with $j_e \perp \hat{m}_{Py}$, the first term in Eq. (4) is zero. As shown in Fig. 2c, $\psi_{polar}$ for the test sample switches with the applied field direction, and also reverses polarity when $\hat{m}$ is switched, which is consistent with the behavior expected from the second term of Eq. (4). The magnitude of $h_{DL}^R$ is estimated to be 25% of the magnitude of $h_{DL}$, or $30 \pm 4$ $Am^{-1}$. This result confirms the generation of a spin current with rotated

spin polarization by the PML. By use of Eq. (2), we estimate

$$\theta^R = \frac{2e}{\hbar}\frac{|Q_\sigma^R|}{|j_e|} = \frac{2e}{\hbar}\frac{\mu_0 M_{Py} d_{Py} h_{DL}^R}{|j_e|} \approx (4.8 \pm 0.6) \times 10^{-3}, \quad (6)$$

under the assumption of perfect spin absorption at the Py/Cu interface, where $\mu_0 M_{Py} = 1$ T and $d_{Py} = 2$ nm are the saturation magnetization and thickness of Py, respectively. For the control sample, where $Q_\sigma^R$ is presumably suppressed by the $TaO_x$ layer, $\psi_{polar}$ is independent of $\hat{m}$. The slight dependence of $\psi_{polar}$ on the applied field is possibly due to misalignment of the applied field and the current flow direction, which is estimated to be about 1.5°.

We decomposed $\psi_{polar}$ into the component that is even in $\hat{m}$ $(\Delta\psi_{polar}^{even})$ and odd in $\hat{m}$ $(\Delta\psi_{polar}^{odd})$, by defining

$$\Delta\psi_{polar}^{even} = \left(\psi_{polar}^{++} - \psi_{polar}^{+-}\right) + \left(\psi_{polar}^{-+} - \psi_{polar}^{--}\right)$$
$$\Delta\psi_{polar}^{odd} = \left(\psi_{polar}^{++} - \psi_{polar}^{+-}\right) - \left(\psi_{polar}^{-+} - \psi_{polar}^{--}\right), \quad (7)$$

where the first superscript in $\psi_{polar}^{++}$ denotes the sign of $\hat{m}$ and the second superscript denotes the sign of $\hat{m}_{Py}$ during the measurement as illustrated in Fig. 2c. We then measured the dependence of $\Delta\psi_{polar}^{even}$ and $\Delta\psi_{polar}^{odd}$ on the applied field angle in the sample plane, where $\phi$ is the angle between the applied field and the charge current direction. As shown in Fig. 2d, $\Delta\psi_{polar}^{even}$ is proportional to $\cos\phi$, whereas $\Delta\psi_{polar}^{odd}$ is proportional to $\sin\phi$, consistent with the phenomenology expressed in Eq. (4).

We also measured $m_{Py}^{\|}$ using the quadratic MOKE response[25], and observed the same dependencies on $\hat{m}$ as for the DL torque. The quadratic MOKE response, $\psi_{quad}$, is proportional to $(\hat{m}_{Py} \cdot \hat{y})(\hat{m}_{Py} \cdot \hat{x})$. When the charge current is parallel to $H_{ext}$ ($\phi = 0°$), as depicted in Figs. 2a and 3a, shows that both the test

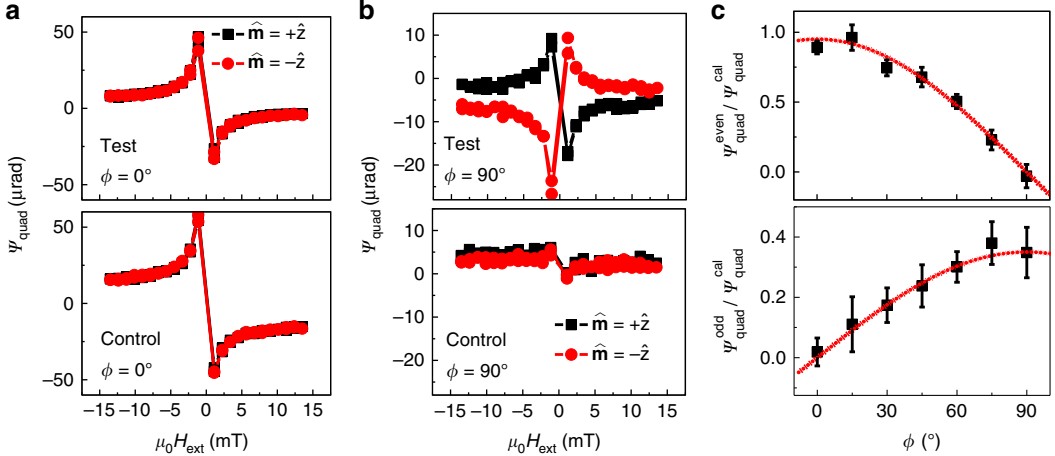

**Fig. 3** Current-induced field-like torque. Here we use quadratic MOKE magnetometry to measure current-induced in-plane effective fields in the same configuration as shown in Fig. 2a. In this measurement, normal incidence light with linear polarization along the x-direction is used. Although in this measurement, the polar MOKE signal is also detected, it is much weaker than the quadratic MOKE response and can be distinguished from fittings. **a** The quadratic MOKE response measured in the test and control samples when current is applied parallel to $H_{ext}$. No dependence on the initial magnetization of the PML, $\hat{m}$, is observed. **b** The quadratic MOKE response measured in the test and control samples when current is applied perpendicular to $H_{ext}$. In the test sample, the quadratic MOKE response is reversed when $\hat{m}$ is reversed. On the contrary, the weak quadratic MOKE response in the control sample, which is likely due to small misalignment of $H_{ext}$, has little dependence on $\hat{m}$. **c** The angle dependence of $\psi_{quad}^{even}$ and $\psi_{quad}^{odd}$ of the test sample. Here the values are extracted by performing a linear fitting with $\psi_{quad}^{cal}$, which is the quadratic MOKE response measured with an external AC calibration field of $117\pm10$ $Am^{-1}$. The slope is extracted and plotted here as $\psi_{quad}^{even}/\psi_{quad}^{cal}$ and $\psi_{quad}^{odd}/\psi_{quad}^{cal}$, respectively. Red lines are fittings with $\psi_{quad}^{even}/\psi_{quad}^{cal} = 0.95 \cos\left(\frac{\pi}{180}\phi\right)$ and $\psi_{quad}^{odd}/\psi_{quad}^{cal} = 0.35 \sin\left(\frac{\pi}{180}\phi\right)$. The error bars represent the standard deviation associated with the linear fitting performed on $\psi_{quad}^{cal}$

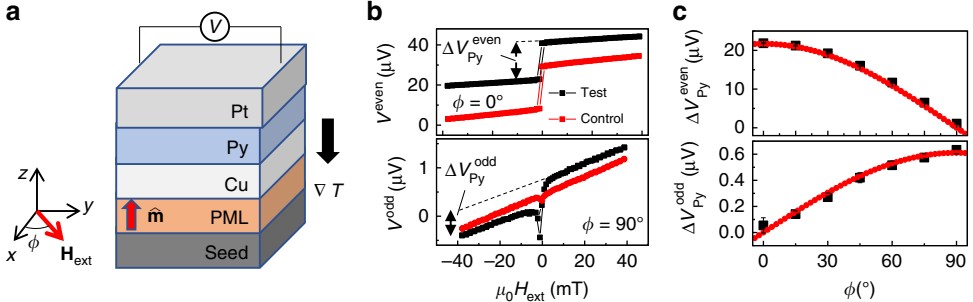

**Fig. 4** Spin Seebeck effect-driven spin galvanic effect measurements. **a** Experimental configuration for the SSE-driven SGE measurements. **b** Voltages measured in the two different configurations for the test sample (seed/PML/Cu/Py/Pt) and control sample (seed/PML/Cu/TaO$_x$/Py/Pt). $V^{\text{even}}$ and $V^{\text{odd}}$ are the sum and difference of the two voltage curves when $\hat{\mathbf{m}}$ is polarized up and down, respectively. **c** Angle dependence of the voltage signals associated with Py switching. Here $\Delta V_{\text{Py}}^{\text{even}}$ is the first term of $V^{\text{even}}$ in Eq. (9) and $\Delta V_{\text{Py}}^{\text{odd}}$ is the first term of $V^{\text{odd}}$ in Eq. (9). The red curves are fits to $\Delta V_{\text{Py}}^{\text{even}} = 21.7 \cos\left(\frac{\pi}{180}\phi\right)$ and $\Delta V_{\text{Py}}^{\text{odd}} = 0.61 \sin\left(\frac{\pi}{180}\phi\right)$. The error bars represent the standard deviation calculated from the linear fitting in **b** used to calculate $\Delta V_{\text{Py}}^{\text{even}}$ and $\Delta V_{\text{Py}}^{\text{odd}}$

and control samples exhibit quadratic MOKE responses that are proportional to $1/H_{\text{ext}}$ but independent of $\hat{\mathbf{m}}$, as expected from Eq. (5). When the charge current is applied perpendicular to $H_{\text{ext}}$ ($\phi = 90°$), as shown in Fig. 3b, a significant quadratic MOKE signal with a sign dependence on $\hat{\mathbf{m}}$ is obtained with the test sample. This is consistent with the second term in Eq. (5). On the other hand, the quadratic MOKE signal for the control sample is independent of $\hat{\mathbf{m}}$, as expected, since $\mathbf{Q}_{\hat{\sigma}}^{\text{R}}$ is blocked by the TaO$_x$ layer. We measured $h_{\text{FL}}^{\text{R}} = 41 \pm 13\,\text{Am}^{-1}$ with the test sample, where we employed the same current (30 mA through a 50 μm strip) as was used in the polar MOKE measurement. As such, the FL torque is comparable in magnitude to the DL torque due to $\mathbf{Q}_{\hat{\sigma}}^{\text{R}}$.

We decompose $\psi_{\text{quad}}$ into components that are either even or odd in $\hat{\mathbf{m}}$ by defining

$$\psi_{\text{quad}}^{\text{even}} = \left[\psi_{\text{quad}}^{+} + \psi_{\text{quad}}^{-}\right]/2$$
$$\psi_{\text{quad}}^{\text{odd}} = \left[\psi_{\text{quad}}^{+} - \psi_{\text{quad}}^{-}\right]/2, \tag{8}$$

where the superscripts + and − denote whether $\hat{\mathbf{m}}$ is oriented along +z or −z, respectively. We further perform linear fittings of $\psi_{\text{quad}}^{\text{even}}$ and $\psi_{\text{quad}}^{\text{odd}}$ with a calibration signal $\psi_{\text{quad}}^{\text{cal}}$ measured with an external calibration field[25], and plot the slopes as a function of $\phi$ in Fig. 3c. As expected from the Eq. (5), $\psi_{\text{quad}}^{\text{even}}$, which presumably results from the sum of $h_{\text{Oe}}^{\parallel}$ and $h_{\text{FL}}$, is proportional to $\cos\phi$; while $\psi_{\text{quad}}^{\text{odd}}$, which is ostensibly the result of $h_{\text{FL}}^{\text{R}}$ generated by $\mathbf{Q}_{\hat{\sigma}}^{\text{R}}$, is proportional to $\sin\phi$.

**SSE-driven spin Galvanic effect measurements**. To further validate our findings, we also measured the spin-rotation symmetry of the SGE, described in Fig. 1b, with a SSE-driven SGE measurement of the same samples. As shown in Fig. 4a, when the samples are subject to an out-of-plane temperature gradient, a spin current is generated due to the SSE[27, 28], which then generates an in-plane voltage. The voltage may arise from the anomalous Nernst effect in the magnetic layers, the SGE due to the spin currents injected into the adjacent layers, as well as the planar Nernst effect[29] in the PML. Depending on whether it has an even or odd symmetry with $\hat{\mathbf{m}}$, the voltage can be described as

$$V^{\text{even}} = \eta_{\text{Py}}\left(\nabla T \times \hat{\mathbf{m}}_{\text{Py}}\right) \cdot \hat{\mathbf{y}} + \eta_{\text{PML}}\left(\nabla T \times \hat{\mathbf{m}}\right) \cdot \hat{\mathbf{y}}$$
$$V^{\text{odd}} = \eta_{\text{R}}\nabla T \times \left(\hat{\mathbf{m}}_{\text{Py}} \times \hat{\mathbf{m}}\right) + \eta_{\text{PML}}^{\text{PNE}}(\hat{\mathbf{m}} \cdot \hat{\mathbf{y}})(\hat{\mathbf{m}} \cdot \nabla T), \tag{9}$$

where $\nabla T$ is the temperature gradient in the z-direction, $\eta_{\text{Py}}$ and $\eta_{\text{PML}}$, with units of V mK$^{-1}$ are the additive anomalous Nernst

and SGE coefficients associated with the Py and PML layers, respectively, $\eta_{\text{PML}}^{\text{PNE}}$ is the coefficient associated with the planar Nernst effect of the PML, and $\eta_{\text{R}}$ is the coefficient associated to the SGE voltage with spin-rotation symmetry described by the second equation of Eq. (3). Note that $\eta_{\text{R}}$ potentially has two competing sources: the spin current generated in Py that diffuses towards the PML, and the spin current generated in PML that diffuses towards the Py.

As shown in Fig. 4b, $V^{\text{even}}$, measured when $H_{\text{ext}}$ is along the x-direction consists of two components: one resembles the hysteretic switching of Py, and a linear slope related to the magnetization tilting of the PML under the influence of external field, as understood from Eq. (6). When $H_{\text{ext}}$ is applied along the y-direction, $V^{\text{even}}$ vanishes. $V^{\text{odd}}$ measured for the control sample yields a straight line, which is consistent with the planar Nernst effect described in Eq. (9). However, $V^{\text{odd}}$ measured for the test sample has an additional component related to the Py magnetization switching, which is consistent with the third term in Eq. (9) due to the SGE with spin-rotation symmetry. Shown in Fig. 4c, the angle dependences of the voltage signal further confirm the symmetry described by Eq. (9).

## Discussion

The sample used in the spin–orbit torque measurement is a spin valve. An in-plane charge current perturbs the electron distribution thus leading to interlayer spin-dependent scattering as observed in the giant magnetoresistance effect[30]. In this process, the spin-dependent scattering may generate a spin transfer torque (STT) on the Py layer that is different from the spin–orbit effects. However, this STT is independent of the in-plane current direction. The MOKE response to the charge current due to the STT is likely to be second order and therefore is not picked up in our detection.

In the discussion above, we imply the SOT with rotated symmetry originates from $\mathbf{Q}_{\hat{\sigma}}^{\text{R}}$ that is generated near the PML/Cu interface. However, $\mathbf{Q}_{\hat{\sigma}}^{\text{R}}$ may also be generated through an alternative process: a spin current is generated from the bulk Py or the Cu/Py and Py/Pt interfaces with conventional symmetry, which diffuses and creates a spin accumulation near the Cu/PML interface. Under the influence of the imaginary part of spin mixing conductance near the Cu/PML interface, this spin accumulation can also generate a spin current with spin-rotation symmetry. In this case, $\mathbf{Q}_{\hat{\sigma}}^{\text{R}}$ is likely to depend on the Py thickness and the capping layer of Py, when Py is thinner than its spin-diffusion length. As shown in the Supplementary Note 4, we have measured the DL torques with spin-rotation symmetry in samples with Ta capping and various Py thicknesses, but found $\mathbf{Q}_{\hat{\sigma}}^{\text{R}}$ to be

nearly independent of Py thickness and the capping layer material. Further analysis based on the magnetoelectronic circuit theory[31] also suggests that this process cannot account for the large signal observed experimentally.

There are two possible artifacts in the experiment that could lead to signals with the same spin-rotation symmetry. For example, the anomalous Hall effect in the PML can generate a new in-plane charge current, which flows perpendicular to the applied charge current. This new charge current can then generate SOT and SGE from the conventional spin–orbit effects, but appears to have spin-rotation symmetry, as the direction of the new charge current depends on the PML magnetization. However, as discussed in the Supplementary Note 5, this contribution should scale with the anomalous Hall angle of the sample, and is estimated to be much weaker than the observed signal.

If there is a magnetostatic coupling between the PML and Py layer, the magnetizations of the two will tilt away from the designated directions. In that case, a current-induced in-plane effective field with conventional symmetry may cause polar MOKE responses that appear to be from the DL torque with spin-rotation symmetry. However, this effect is estimated to be very weak in our sample as discussed in Supplementary Note 6.

It should be pointed out that the SOE with spin-rotation symmetry may not only arise from the interface between the very top layer of the PML and Cu. The PML consists of many interfaces of $Ni/Co_{90}Fe_{10}$, which are known to have a strong spin–orbit interaction that gives rise to the perpendicular anisotropy. As each layer in the PML is very thin, the observed SOE may partially arise from the $Ni/Co_{90}Fe_{10}$ interfaces within the PML.

The SOE with spin-rotation symmetry in combination with the SOE with conventional symmetry can generate spin current with arbitrary polarization simply by adjusting the magnetization direction. An important implication of these findings is the ability to generate a perpendicular polarized spin current by use of a FM/NM interface, where the FM is magnetized collinear to the current flow direction. Such a spin-current polarization is required to switch a perpendicular magnetized layer by use of anti-damping STT alone. These findings can significantly benefit the development of MRAM technology, where perpendicularly magnetized memory is more favorable as it allows for high stability and scalability[32]. Although the verification of the spin-rotation phenomenology presented here does not permit us to predict the efficiency of the perpendicularly polarized spin-current generation for the case of collinear current and FM magnetization, we think the key to high efficiency is through interface optimization, where spin–orbit interaction, spin precession and dephasing should all be taken into account.

## Methods

**Sample Fabrication.** The samples used in this study are fabricated by magnetron sputtering. The $TaO_x$ layers in the control samples are made by depositing 1.5 nm Ta film and subsequently exposing to the air. This process is repeated to fabricate a total of 3 nm of $TaO_x$.

**MOKE magnetometry measurement.** In the MOKE measurement, the total charge current applied is 30 mA, from which we estimate the current density through PML to be about $3.8 \times 10^{10}$ $Am^{-2}$. The principle of the SOT detection with MOKE magnetometry and detailed protocols can be found in reference[25]. In the measurement, we apply a small in-plane sweeping external magnetic field $H_{ext}$ (<15 mT) that aligns the magnetization of Py. Owing to the large anisotropy (~390 mT), the magnetization of the PML remains mostly perpendicular when $H_{ext}$ sweeps in the film plane. We set the initial magnetization direction of the PML by placing a permanent magnet close to the sample and then remove it. The permanent magnet generates about 50 mT field perpendicular to the film plane, whereas the coercivity of the PML is about 30 mT. We measure the hysteretic loops 10–20 times and take the average.

**Thermal measurement.** In the thermal measurement, the samples are typically cut into 2 mm × 25 mm strips. The voltages across the samples are measured by a

Keithley nano voltmeter 2182. The samples are sandwiched between two aluminum plates. The aluminum plates are attached to Peltier elements to create a temperature difference across the sample. The typical temperature difference, $\Delta T$, measured on the two aluminum plates is about 50 K. All voltages are scaled to a 50 K temperature difference by taking $V/\Delta T \times 50$. Similar to the MOKE measurement, we switch the magnetization of PML by placing a permanent magnet close to the sample and then removing it. We measure the hysteretic loops 10–20 times and take the average. Possible drifts in the measurement are subtracted by assuming the drift is linear with measurement time.

**Data availability**. The data that support the findings of this study are available from the corresponding author upon request.

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

## Acknowledgements

The work done at the University of Denver is partially supported by the PROF and the Partners in Scholarship grants and by the National Science Foundation under Grant Number ECCS-1738679. The work done at University of Delaware is supported by NSF DMR1505192. We would like to thank Vivek Amin, Mark Stiles, Satoru Emori, and Barry Zink for critical reading of the manuscript and illuminating discussions. We would also like to thank Wenrui Wang for help with MOKE measurements.

## Author contributions

X.F. conceived and designed the experiments; E.R.J.E. and J.M.S. fabricated the samples; E.R.J.E., S.R.A., and T.W. characterized the samples; A.M.H., S.R.A., and X.F. performed the thermal measurement; T.W. patterned the samples and performed the MOKE measurements. All authors contributed to analysis and interpretation of the data.

## Additional information

**Competing interests:** The authors declare a competing financial interest. A patent application related to this research has been filed.

