## [Peer Review File · Nature Communications]

Reviewers' comments:

Reviewer #1 (Remarks to the Author):

The paper Observation of Spin Orbit Effects with spin rotation symmetry reports on a novel spin orbit interaction force at the interface between a heavy metal material and a ferromagnetic material, based on the spin Orbit effect, whereby a spin current generated via a charge current experiences a rotation of the spin polarization at the interface due to precession about the magnetization similar to the rotation of polarization of light. The manuscript presents experimental results to demonstrate the existence of such spin rotation and also its corresponding inverse effect using a specific multilayer structure incorporating two ferromagnetic layers of orthogonal magnetization. The results potentially will have important implications and provide a means to expand the ways of manipulation of magnetization by spin orbit torques. The idea of the experiments and analysis of the results are convincing by comparing two types of structures (one control device for which the spin current with rotated symmetry is blocked off). However, before publication the manuscript has to clarify the points listed below. In particular the presentation of the different results are very short and details of the explanations and/or derivations are either missing or have to be searched in the figure caption or the very lengthy supplemental material. This makes the reading and understanding of the results difficult for a non-specialist reader but also does not allow a straightforward assessment of the paper to rule out that other effects or interpretation of the results are possible. Therefore, it is strongly suggested that the authors present their results in a more transparent way that makes it easier to follow their reasoning. Also, it is very much regretted that a lot of material is left to the supplemental (which is longer than the actual paper) which somewhat suggests that Nature Communications might not be the adequate journal to publish the work.

1) Fig. 2a, the schematics of the experimental measurement configuration should indicate more clearly details: the sample structure (micron sized device), what is the MOKE configuration (plane of incidence, polarization of light), field orientation and current flow as well as the Oersted field. Eq. 4 suggests an out of plane component h_{0e}^{\perp} that should be important only for small sized structures and not for micronsized devices. This terms needs explanation

2) For the experimental demonstration discussed in Fig. 2, a magnetic stack of PML/Cu/Py/Pt (PML = perpendicular material) is used with Py being the spin current detector. Its static equilibrium orientation is probed via the Magneto optic Kerr effect. Two spin currents are considered whose corresponding torques reorient the magnetization of the Py layer: the 'standard' Spin Hall effect at the Pt/Py interface and the PML/Cu interface and spin current of rotated polarization arising at the PML/Cu interface. The choice of spin currents considered are not well explained and motivated. In total there are three interfaces that can give rise to a spin current Pt/Py, Cu/Py and PML/Cu. Why is only the PML/Cu interface considered and not the Py/Cu interface, what are the supports that a considerable spin current is generated using Cu as non-magnetic material, which is not usually considered for such SOT experiments. Why is a rotation of spin polarization considered to occur only at the PML/Cu interface (at least this has to be inferred from the text on the fifth page where the rotated spin polarization is given as $\sigma//(\mathbf{m} \times (\mathbf{j}_e \times \mathbf{z}))$. This is not clearly expressed in the text itself). Why is there no spin rotation at the Pt/Py or Py/Cu interface, which should occur according to the definition given in Fig. 1a of the effect of rotation of spin polarization. The manuscript, nor the supplemental material provide any explanation on the choice of spin currents and the spin polarization considered.

3) Similarly, the derivation of equation 4 is not very transparent, although it is referred to section 3 of the supplemental, there is no direct link for instance between equation 4 and equations S2-S4. Is m_{py}^{\perp} given by $\delta\theta$? Please provide a definition of the magnetization and spherical coordinate system and a better link between the equations in the main text and supplemental.

4) Fig. 2 discusses results that are analyzed using equation 4 that only considers the damping like torque, which results in an out-of-plane component of m_{py} . To distinguish the contributions from the

standard term and the one with rotated spin polarization, the angle between field and current is varied and the fits are shown in Fig. 2d. It takes a while to understand what is plotted in Fig. 2d, the definition being left to the figure caption. It would be helpful to indicate for instance on Fig. 2c top, the black points in positive field correspond to ψ^{++} , and so on for the other branches (red and black points in positive and negative field). At least this is supposed of how the even and odd values of $\Delta\psi^{\text{even/od}}$ were deduced vs angle. In Figure caption 2d it should be explained better what is meant by "Values when m_{Py} is saturated are used in the extrapolation". Does m_{Py} not saturate in the field range at different angles, so that an extrapolation becomes necessary. If so why does it not saturate?

5) For the extraction of the strength of the direct torque field and the one of rotated polarization, an assumption of the current density is made. On page 6, first paragraph it is said the "by performing a linescan, we estimate..." Here it is not at all clear what the line scan is referring to, without having to look up a reference, nor why the linescan is necessary to deduce h_{DL} . It is not asked to give any details, but at least to provide some explanation that motivates and makes it plausible. In the same context it is not clear what is meant by the integrated current density, integrated over the stack thickness? If so why is for estimating h_{DL} the integrated current used, while for h_{DL} the current density in PML is used. Cannot the same estimate be made to determine the current density in Pt.

6) Has the effect been tested as a function of the thickness of the Py layer or for different spacer materials.

Reviewer #2 (Remarks to the Author):

This paper presents convincing experimental evidence for new spin-polarization effects associated electron spin interaction with thin ferromagnetic layers. The authors study multilayers that consist of a perpendicularly magnetized metallic layer (a Co/Ni multilayer), a non-magnetic layer (Cu) and a in-plane magnetized NiFe (Py) layer. Through measurement of the spin-torques on a NiFe (Permalloy) layer with current flowing in the plane of the layers, the authors deduce that there is a spin-polarization component in the direction of the electric current flow (in contrast to the typical spin-Hall effect case in which the spin-polarization is in the plane but perpendicular to the direction of current flow.)

The authors present results on control samples in which a thin insulating layer (TaOx) interrupts the spin-current flow associated with the electron spin interaction with the perpendicularly magnetized layer, showing the torque effect is the conventional one when the insulating layer is present.

They also present spin-Seebeck effect measurements that show a symmetry consistent with the spin-torque measurements. Further their supplementary materials discuss a number of alternative origins of the effect and provide evidence that these effects cannot explain their observations or would be smaller in magnitude than the effects they observe.

In sum, I feel this paper will generate considerable interest and stimulate further work--particularly, as noted by the authors, on generating a perpendicularly polarized spins to switch magnets with perpendicular magnetization. I thus believe the paper is suitable for publication in Nature Communications, nearly as is.

I suggest minor clarifications on the nature of the spin-currents generated. There are several places in the manuscript in which the authors refer to "two spin currents". Persummable there is a well-defined (average) direction of spin-polarization of the ensemble of diffusing spins. Therefore, I suggest the authors make clear that the spin-polarization is a superposition of the spin-currents in-plane and parallel to the current and in-plane and perpendicular to the current in any given sample.

Reviewer #3 (Remarks to the Author):

The Authors claim to have found a new type of spin orbit effect (SOE) which is able to apply a (perhaps tunable) arbitrary spin-orbit-torque (SOT) on a ferromagnet. This is sketched as an ideal candidate for future MRAM as this would eliminate the use of other symmetry breaking effects to guarantee full 180 degree deterministic switching. This is certainly a very relevant issue and could give rise to a new direction for SOT based devices.

I however find the presented experimental results and explanation unsatisfactory. The theory is based on a phenomenological equations which could describe many SOT and STT observations, no in-depth discussion is spent on the exact mechanism which can generate the alleged torque. Where is it generated, what is the role of all interfaces, what is the interplay. Moreover, the used stack to prove the mechanisms contains both a heavy metal Pt layer (conventionally the source of SOT) and a PMA multilayer stack consisting of Co/Ni repeats. Hence, many many effects can be at the origin of spin-transfer-torques and SOT's, and a intricate interplay can cause a multitude of observations. An extensive systematic study of layer thicknesses, interface engineering(s) and significant statistics would be needed to convince me of such a phenomenological claim.

The authors are concentrating a lot of effort on what the observations cannot be at the origin of the observed effect in the supplementary sections, albeit I respect the authors that they have gone such a mile in working out the details. I miss simple experimental support of these points. For instance on simple static coupling issues. It is much more convincing to show magnetometry data of the stack along all orthogonal directions, if done carefully any coupling mechanism (Orange Peel, RKKY or pin hole, etc..) will directly show up. Furthermore, why is the Pt on top of the Py not discussed as a source of SOT, any why does this not show up in the experiments? Why is a possible STT from the bottom PMA stack not discussed. Also simple experimental details of how the 50x50um stacks are contacted and how they are mounted in the experimental setup are missing. Is the current really flowing unidirectionally through the 50x50um dot, something I cannot assess from the presented support? There is a loose remark somewhere on a misalignment issue, addressing such issues carefully and systematically are paramount in addressing the symmetries of these torques and providing proof for the claim.

Considering these major issues I cannot recommend the manuscript. Major effort should be spent on basic systematic studies i.e. layer thicknesses, current densities, interface engineering and extensive magnetometry to give more confidence on the basics of the stack and what the effects are on the observed signal.

Yours,
Reinoud Lavrijsen

Review 1

1) Fig. 2a, the schematics of the experimental measurement configuration should indicate more clearly details: the sample structure (micron sized device), what is the MOKE configuration (plane of incidence, polarization of light), field orientation and current flow as well as the Oersted field. Eq. 4 suggests an out of plane component h_{Oe}^\perp that should be important only for small sized structures and not for micron sized devices. This term needs explanation

We thank the reviewer for the suggestions to make the paper easier to read. Figure 2 is now modified to include the details of the sample, field and current orientation as well as diagrams with effective fields. The out-of-plane Oersted field does not depend on magnetization orientations and averages out to be zero in the whole sample. Its contribution to the magnetization reorientation is trivial. But its magnitude may not be small even for micron sized device comparing to other current-induced effective fields. In fact, we use the spatially varying out-of-plane Oersted field to calibrate the MOKE response (Nat. Comm. 5, 3042 (2014)).

2) For the experimental demonstration discussed in Fig. 2, a magnetic stack of PML/Cu/Py/Pt (PML = perpendicular material) is used with Py being the spin current detector. Its static equilibrium orientation is probed via the Magneto optic Kerr effect. Two spin currents are considered whose corresponding torques reorient the magnetization of the Py layer: the ‘standard’ Spin Hall effect at the Pt/Py interface and the PML/Cu interface and spin current of rotated polarization arising at the PML/Cu interface. The choice of spin currents considered are not well explained and motivated. In total there are three interfaces that can give rise to a spin current Pt/Py, Cu/Py and PML/Cu. Why is only the PML/Cu interface considered and not the Py/Cu interface, what are the supports that a considerable spin current is generated using Cu as non-magnetic material, which is not usually considered for such SOT experiments. Why is a rotation of spin polarization considered to occur only at the PML/Cu interface (at least this has to be inferred from the text on the fifth page where the rotated spin polarization is given as $\vec{Q}_\sigma^R // (\mathbf{m} \times (\mathbf{j}_e \times \mathbf{z}))$. This is not clearly expressed in the text itself). Why is there no spin rotation at the Pt/Py or Py/Cu interface, which should occur according to the definition given in Fig. 1a of the effect of rotation of spin polarization. The manuscript, nor the supplemental material provide any explanation on the choice of spin currents and the spin polarization considered.

We thank the reviewer for raising this important question. Spin current with spin rotated around Py magnetization acting back on Py will give rise to a damping-like torque that exhibits the same symmetry as a field-like torque with conventional symmetry, and a field-like torque that exhibits the same symmetry as a damping-like torque with conventional symmetry. Therefore, we consolidate all the contributions from the Pt/Py and Cu/Py interfaces to h_{FL} and h_{DL} . But in the previous version of manuscript, this was not explained explicitly. We have rewritten the following part to introduce all current-induced effective fields

“According to Eq. (2), an in-plane charge current \vec{j}_e generates spin currents with three components that exert torques on the Py magnetization $\hat{m}_{Py} : \vec{Q}_\sigma$ with $\hat{\sigma} // (\hat{j}_e \times \hat{z})$ due to the spin-orbit effect (SOE) with conventional symmetry near the Pt/Py and PML/Cu interfaces, \vec{Q}_σ^R

with $\hat{\sigma} \parallel \hat{m} \times (\hat{j}_e \times \hat{z})$ due to the SOE with spin rotation symmetry near the PML/Cu interface and $\bar{q}_{\hat{\sigma}}^R$ with $\hat{\sigma} \parallel \hat{m}_{\text{Py}} \times (\hat{j}_e \times \hat{z})$ due to the SOE with spin rotation symmetry near the Cu/Py and Py/Pt interfaces. Here, \hat{m} is the unit magnetization vector of the PML and \hat{j}_e is the unit vector along the direction of the applied current. In general, a spin current with spin polarization $\hat{\sigma}$ can exert two types of spin torques on the Py magnetization: a damping-like (DL) torque in the direction of $(\hat{m}_{\text{Py}} \times \hat{\sigma}) \times \hat{m}_{\text{Py}}$, which is equivalent to an effective field in the direction of $\hat{m}_{\text{Py}} \times \hat{\sigma}$, and a field-like (FL) torque in the direction of $\hat{\sigma} \times \hat{m}_{\text{Py}}$, which is equivalent to an effective field in the direction of $\hat{\sigma}$. Therefore, there are four effective fields due to the various spin-orbit effects that act on the Py magnetization: $h_{\text{DL}} \hat{m}_{\text{Py}} \times (\hat{j}_e \times \hat{z})$ due to the damping-like torque from $\bar{Q}_{\hat{\sigma}}$ and the field-like torque from $\bar{q}_{\hat{\sigma}}^R$, $h_{\text{FL}} (\hat{j}_e \times \hat{z})$ due to the field-like torque from $\bar{Q}_{\hat{\sigma}}$ and the damping-like torque from $\bar{q}_{\hat{\sigma}}^R$, and $h_{\text{DL}}^R \hat{m}_{\text{Py}} \times [\hat{m} \times (\hat{j}_e \times \hat{z})]$ and $h_{\text{FL}}^R \hat{m} \times (\hat{j}_e \times \hat{z})$ due to the damping-like and field-like torques from $\bar{Q}_{\hat{\sigma}}^R$ respectively. It should be emphasized the possible effective fields due to the Rashba-Edelstein spin-orbit effects at the Cu/Py and Pt/Py interfaces will share the same symmetry as $h_{\text{DL}} \hat{m}_{\text{Py}} \times (\hat{j}_e \times \hat{z})$ and $h_{\text{FL}} (\hat{j}_e \times \hat{z})$, thus not discussed separately.

Besides the SOTs, an in-plane current also generates a uniform in-plane Oersted field h_{Oe}'' , and a spatially-varying out-of-plane Oersted field h_{Oe}^+ .

In our hypothesis, the spin currents that are proportional to the PML magnetization are either generated within the PML itself or at the PML/Cu interface. Because of strong dephasing effects within the bulk of ferromagnetic metals, the effective spin current source is likely to be located very close to the surface of the PML even if the spin current is generated within the bulk of the PML. Therefore, we use the expression that presumes the spin current is generated at the PML/Cu interface in order to generalize the possible origin of the spin current. Indeed, the interface between Cu and a ferromagnetic metal is often neglected in previous studies, but in our experiment, we must use Cu as the spacer, which allows spin current to flow through. The fact that we observe spin-orbit effects depending on the PML magnetization suggests either that the spin current is generated solely by PML or that the PML/Cu interface has some non-negligible spin-orbit interactions.

3) Similarly, the derivation of equation 4 is not very transparent, although it is referred to section 3 of the supplemental, there is no direct link for instance between equation 4 and equations S2-S4. Is m_{Py}^{\perp} given by $\delta\theta$? Please provide a definition of the magnetization and spherical coordinate system and a better link between the equations in the main text and supplemental.

We thank the reviewer for the suggestion. In the Supplementary Information, we have defined the magnetization and spherical coordinates in Fig. S2. m_{Py}^{\perp} given by $\delta\theta$ and m_{Py}'' given by $\delta\phi$ are

also shown in Eq. (S3). With the introduction of all effective fields, we think the derivation of Eq. (4) is clearer than the previous version.

4) Fig. 2 discusses results that are analyzed using equation 4 that only considers the damping like torque, which results in an out-of-plane component of m_{Py} . To distinguish the contributions from the standard term and the one with rotated spin polarization, the angle between field and current is varied and the fits are shown in Fig. 2d. It takes a while to understand what is plotted in Fig. 2d, the definition

being left to the figure caption. It would be helpful to indicate for instance on Fig. 2c top, the black points in positive field correspond to ψ^{++} , and so on for the other branches (red and black points in positive and negative field). At least this is supposed of how the even and odd values of $\psi^{\text{even/odd}}$ were deduced vs angle. In Figure caption 2d it should be explained better what is meant by “Values when m_{Py} is saturated are used in the extrapolation”. Does m_{Py} not saturate in the field range at different angles, so that an extrapolation becomes necessary. If so why does it not saturate?

We have labelled ψ^{++} and other three values in Fig. 2c. As for the “Values when m_{Py} is saturated are used”, this was meant to refer to that we always use MOKE values measured at large field ($\sim \pm 100$ Oe) rather than those around 0 Oe. For Py , the magnetization is always saturated in all measurement directions. To avoid the confusion, and because ψ^{++} is already labelled in Fig. 2c, we deleted that sentence from the figure caption.

5) For the extraction of the strength of the direct torque field and the one of rotated polarization, an assumption of the current density is made. On page 6, first paragraph it is said the “by performing a linescan, we estimate...” Here it is not at all clear what the line scan is referring to, without having to look up a reference, nor why the linescan is necessary to deduce h_{DL} . It is not asked to give any details, but at least to provide some explanation that motivates and makes it plausible. In the same context it is not clear what is meant by the integrated current density, integrated over the stack thickness? If so why is for estimating h_{DL} the integrated current used, while for h_{DL}^R the current density in PML is used. Cannot the same estimate be made to determine the current density in Pt.

We have added some description to show why a linescan measurement can be used for calibration.

“The Oersted field, h_{Oe}^\perp , is independent of \hat{m} and \hat{m}_{Py} and spatially varies transverse to \vec{j}_e . We use h_{Oe}^\perp , calculated by use of the Biot-Savart Law, to calibrate the magnitude of h_{DL} and h_{DL}^R , both of which depend on \hat{m}_{Py} and are uniform across the sample. We scan the laser across the sample in the direction perpendicular to the applied current and isolate the polar MOKE response that is independent to \hat{m}_{Py} . This component of polar MOKE signal is proportional to h_{Oe}^\perp and its spatial distribution is fitted with the Biot-Savart Law²⁶. This fitting is used to establish the sensitivity of the polar MOKE measurements, i.e. we determine the ratio of the polar MOKE signal and the corresponding out-of-plane field, which can then be used to evaluate h_{DL} and h_{DL}^R in subsequent measurements.”

We have changed the description of current density as

“From the calibration, we determined that h_{DL} is 120 ± 12 A/m for a current densities of 1.2×10^{10} A/m² in the Pt layer and 3.8×10^{10} A/m² in the PML. ” If we assume h_{DL} arises solely from Pt, we can estimate the Pt to have an effective spin Hall angle of about 0.06, close to what is typically reported from spin-orbit torque measurements. However, we think h_{DL} also arises in part from conventional spin current generation at the PML. As a result, such an estimate of the spin Hall angle in Pt is subject to substantial systematic errors and is only provided here to verify that the measured magnitude of h_{DL} is within credible bounds.

6) Has the effect been tested as a function of the thickness of the Py layer or for different spacer materials.

We have measured the spin-orbit torques with conventional and spin rotation symmetries as a function of Py thicknesses. The data is summarized in the Supplementary Information Figure S3 (c) and (d). Both spin-orbit torques scale inversely with the Py thickness. We have studied different spacer materials in a system with Pt/Co bilayer as the PML. However, we plan to disseminate those results in a separate publication.

Review 2

I suggest minor clarifications on the nature of the spin-currents generated. There are several places in the manuscript in which the authors refer to "two spin currents". Presumably there is a well-defined (average) direction of spin-polarization of the ensemble of diffusing spins. Therefore, I suggest the authors make clear that the spin-polarization is a superposition of the spin-currents in-plane and parallel to the current and in-plane and perpendicular to the current in any given sample.

We thank the reviewer for the positive evaluation. Instead of using “two spin currents”, we now refer to the spin currents with conventional symmetry and spin rotational symmetry as “two components of spin currents” throughout the manuscript. For example,

“As shown in Fig. 1(a), when an in-plane charge current passes through a FM/NM interface, an out-of-plane propagating spin current can be generated with two components in accordance with both Eqs. (1) and (2).”

and

“According to Eq. (2), an in-plane charge current \vec{j}_e generates spin currents with three components that exert torques on the Py magnetization \hat{m}_{Py} ”

Review 3

1. I however find the presented experimental results and explanation unsatisfactory. The theory is based on a phenomenological equations which could describe many SOT and STT observations, no in-depth discussion is spent on the exact mechanism which can generate the alleged torque. Where is it generated, what is the the role of all interfaces, what is the interplay.

In this paper, our goal is to present the discovery of a spin-orbit phenomenology with novel symmetry. Our key finding is the discovery that the interaction of spin-orbit effects and magnetism adds an additional means to control spin current generation. From the clearly delineated phenomenology presented here, it appears there exists a microscopic process of unknown origin by which a transverse spin accumulation precesses around the magnetization at a FM/NM interface. Therefore, from a pure symmetry-based argument, spin-orbit effects with spin rotation symmetry must exist. Of course, conventional wisdom holds that a transverse spin accumulation is subject to rapid spin dephasing at a NM/FM interface, but that wisdom is based on the neglect of strong interfacial spin-orbit effects. Our experiment demonstrates, surprisingly, these effects are not at all negligible in this particular material system.

It is unclear whether the spin precession may take place in the bulk of ferromagnetic metal, near the interface, or through a more sophisticated coupling between magnetization and spin-orbit interactions. However, considering the ongoing controversy as to whether the spin Hall effect or the interface Rashba-Edelstein effect are the source of spin-orbit torques with conventional symmetry (still debated 6 years after the observation of spin-orbit torques in magnetic bilayers), we think it will take a great effort from us and other scientists to understand the exact mechanism of spin-orbit effects with spin rotation symmetry. Therefore, we provide here a convincing demonstration of the novel phenomenological features, but we leave the determination of the microscopic mechanism as an open question for future studies.

2. An extensive systematic study of layer thicknesses, interface engineering(s) and significant statistics would be needed to convince me of such a phenomenological claim.

In the Supplementary Information 4 we have added the thickness dependent study of the free layer Py. We have found the damping-like torques with both conventional symmetry and spin rotation symmetry to be inversely proportional to the Py thickness. This suggests the spin current with spin rotation symmetry is generated from layers other than the Py film itself, consistent with our phenomenological argument.

While the thickness-dependent study of the PML and interface engineering are important for understanding the underlying mechanism, we think those should be follow-up studies and will take a very long time. In this paper, our goal is to report the observation of spin-orbit

effects of a previously unreported symmetry. Through two complementary experiments (spin-orbit torque and spin galvanic effect) and proper control measurements, we think we have presented reasonable evidence to support the phenomenology. Hopefully the additional Py thickness dependence study instills more confidence in the reviewer.

For instance on simple static coupling issues. It is much more convincing to show magnetometry data of the stack along all orthogonal directions, if done carefully any coupling mechanism (Orange Peel, RKKY or pin hole, etc..) will directly show up.

We thank the reviewer for the suggestions. We have performed full magnetic hysteresis loop measurements using vibrating sampling magnetometer, as now shown in Fig. S1. We find the magnetic hysteresis for PML are very close in both the test and control samples, which suggests a weak interlayer coupling in the test sample. We modify the description in the Supplementary Information 2 to describe this measurement:

“The full magnetic hysteresis of the test sample, seed/PML/Cu(3)/Py(2)/Pt(3), and control sample, Seed/PML/Cu(3)/TaO_x(3)/Py(2)/Pt(3), are measured by using vibrating sampling magnetometer, as shown in Fig. S1. For the out-of-plane measurement configuration (Fig. S1 (a)), the sharp switchings at lower fields correspond to the PML magnetization while the slope that saturates gradually at higher fields correspond to the Py magnetization. In both test and control samples, the coercivity of PML are identical but the saturation fields of Py are not the same. The saturation field of Py in the control sample is much lower than that in the test sample, which may be due to the increased perpendicular anisotropy at the TaO_x/Py interface compared to that at the Cu/Py interface. In the in-plane measurement configuration, as shown in Fig. S1 (b), the two hysteresis nearly overlap each other. Due to the thick and insulating spacer layers (Cu(3)/TaO_x(3)) in the control sample, the interlayer coupling in the control sample should be negligible. Since the PML magnetization hysteresis in the test sample behaves very closely to that in the control sample, we think the interlayer coupling in the test sample is not significant either.”

However, we do not think we can extrapolate the interlayer coupling strength with a good accuracy based on the magnetometry data alone. We think the strength of the interlayer coupling can be measured by layer sensitive magnetometry, e.g. XMCD or magnetic tunnel junctions, which however, are not immediately available. But we would like to emphasize that although we have not evaluated the actual strength of the interlayer coupling, we have provided sufficient justification that it cannot be the source for the signal we have observed, as detailed in the Supplementary Information 6.

Furthermore, why is the Pt on top of the Py not discussed as a source of SOT, any why does this not show up in the experiments? Why is a possible STT from the bottom PMA stack not discussed.

We thank the reviewer for pointing out the contribution from Pt, which we should discuss in more details. Pt indeed can be a source of SOT, but will only generate spin-orbit torques with conventional symmetry. In the revised manuscript, we added a paragraph that describe the contributions from all possible layers.

“According to Eq. (2), an in-plane charge current \vec{j}_e generates spin currents with three components that exert torques on the Py magnetization \hat{m}_{Py} : $\vec{Q}_{\hat{\sigma}}$ with $\hat{\sigma} \parallel (\hat{j}_e \times \hat{z})$ due to the SOE with conventional symmetry near the Pt/Py and PML/Cu interfaces, $\vec{Q}_{\hat{\sigma}}^{\text{R}}$ with $\hat{\sigma} \parallel \hat{m} \times (\hat{j}_e \times \hat{z})$ due to the SOE with spin rotation symmetry near the PML/Cu interface and $\vec{q}_{\hat{\sigma}}^{\text{R}}$ with $\hat{\sigma} \parallel \hat{m}_{\text{Py}} \times (\hat{j}_e \times \hat{z})$ due to the SOE with spin rotation symmetry near the Cu/Py and Py/Pt interfaces. Here, \hat{m} is the unit magnetization vector of the PML and \hat{j}_e is the unit vector along the direction of the applied current. In general, a spin current with spin polarization $\hat{\sigma}$ can exert two types of spin torques on the Py magnetization: a damping-like (DL) torque in the direction of $(\hat{m}_{\text{Py}} \times \hat{\sigma}) \times \hat{m}_{\text{Py}}$, which is equivalent to an effective field in the direction of $\hat{m}_{\text{Py}} \times \hat{\sigma}$, and a field-like (FL) torque in the direction of $\hat{\sigma} \times \hat{m}_{\text{Py}}$, which is equivalent to an effective field in the direction of $\hat{\sigma}$. Therefore, there are four effective fields due to the various spin-orbit effects that act on the Py magnetization: $h_{\text{DL}} \hat{m}_{\text{Py}} \times (\hat{j}_e \times \hat{z})$ due to the damping-like torque from $\vec{Q}_{\hat{\sigma}}$ and the field-like torque from $\vec{q}_{\hat{\sigma}}^{\text{R}}$, $h_{\text{FL}} (\hat{j}_e \times \hat{z})$ due to the field-like torque from $\vec{Q}_{\hat{\sigma}}$ and the damping-like torque from $\vec{q}_{\hat{\sigma}}^{\text{R}}$, and $h_{\text{DL}}^{\text{R}} \hat{m}_{\text{Py}} \times [\hat{m} \times (\hat{j}_e \times \hat{z})]$ and $h_{\text{FL}}^{\text{R}} \hat{m} \times (\hat{j}_e \times \hat{z})$ due to the damping-like and field-like torques from $\vec{Q}_{\hat{\sigma}}^{\text{R}}$ respectively. It should be emphasized the possible effective fields due to the Rashba-Edelstein spin-orbit effects at the Cu/Py and Pt/Py interfaces will share the same symmetry as $h_{\text{DL}} \hat{m}_{\text{Py}} \times (\hat{j}_e \times \hat{z})$ and $h_{\text{FL}} (\hat{j}_e \times \hat{z})$, thus not discussed separately.

Besides the SOTs, an in-plane current also generates a uniform in-plane Oersted field h_{Oe}'' , and a spatially-varying out-of-plane Oersted field h_{Oe}^{\perp} .”

The sample used in the measurement is a spin valve. In the current-in-plane configuration, interlayer spin-dependent scattering leads to the observation of the giant magnetoresistance. This in principle also suggests a spin angular momentum transfer or spin transfer torque (STT) between the different magnetic layers that is independent of the spin-orbit effects. However, we note this STT effect is independent of the applied current direction. In our measurement, we apply an alternating current through the sample and detect the first-order

response using a lock-in amplifier. Therefore, the STT effect, which will cause a second-order response, is not measured. We thank the reviewer for pointing out the STT effect. Though we do not think it alters our conclusion, it is certainly an interesting topic that merits a thorough study later. In the main text, we have added a short paragraph discussing the STT effect.

“The sample used in the measurement is a spin valve. An in-plane charge current perturbs the electron distribution thus leads to interlayer spin-dependent scattering as observed in the giant magnetoresistance effect. In this process, the spin-dependent scattering may generate a spin transfer torque (STT) on the Py layer that is different from the spin-orbit effects. However, this STT is independent to the in-plane current direction. The MOKE response to the charge current due to the STT is likely to be second order and therefore is not picked up in our detection.”

Also simple experimental details of how the 50x50um stacks are contacted and how they are mounted in the experimental setup are missing. Is the current really flowing unidirectionally through the 50x50um dot, something I cannot assess from the presented support?

A sketch of the sample pattern is displayed in Figure 2 (a) in the revised manuscript. The contact pads are 200 nm-thick gold and are extended out to be about 1mm in lateral sizes. The connection to the pad is made by indium dot, which may be slightly asymmetric. But we think the high contrast of conductance between gold and the sample should automatically correct the current flow and make it uniform through the sample. This pattern is used in our previous publication (Nat. Comm. 5, 3042 (2014)). In the supplementary information of that publication, we were able to map out the spatial distribution of the out-of-plane Oersted field across the sample, which indicates the current flow should be reasonably uniform. Even if the flow of the current is nonuniform, it cannot explain why the SOT with spin rotation symmetry depends on the PML magnetization.

There is a loose remark somewhere on a misalignment issue, addressing such issues carefully and systematically are paramount in addressing the symmetries of these torques and providing proof for the claim.

In the SOT measurement, our sample is taped to a sample stage that is placed in an electromagnet and in front of an objective. Unfortunately due to the crowded space of the setup, we cannot add a rotational stage to accurately tune the sample angle. However, we did make significant effort to align the sample visually. Indeed, misalignment from the nominal 90° angle configuration by an angle $\delta\phi$ will give rise to a signal proportional to $\sin(\delta\phi)$. Therefore, we estimate $\delta\phi$ to be less than 1.5° by use of the control sample with a TaO_x insert. The distinct feature of the spin-orbit effects with spin rotation is the dependence on the PML

magnetization. Even if there may be a small deviation of the charge current direction from the nominal direction, the measurements when the PML is polarized up and down are subject to the same deviation. Therefore, the dramatic difference between the MOKE/SGE responses when the PML is polarized up and down can only be related to the spin-orbit effects with spin rotation, not sample misalignment. Hence we have no indication that reduction of $\delta\phi$ below 1.5° will alter our basic conclusions.

REVIEWERS' COMMENTS:

Reviewer #1 (Remarks to the Author):

The authors have taken all comments into account and have responded to them with satisfaction. The changes made to the manuscript helped to present the results in a much clearer manner. The referee has no further direct comments, but one question. The results will certainly initiate studies by other groups. While the authors used an optical technique to extract information on the magnetization of the Py layer, covered by a Pt layer, this is not useful for technological applications. Can the authors comment on how the stack can be implemented to readout the signal by electrical means.

Reviewer #3 (Remarks to the Author):

The Authors have addressed my concerns adequately, I appreciate that the manuscript now concentrates more on the observation and all interfaces / materials are now tacked in the main text. Moreover, the main claims are now better postulated/motivated and left for the readers to analyse/iterate on.

I agree that the further experiments (interface engineering, spacer material variations, etc) will be intensive work and not trivial. Furthermore, it will not support the core message of this work; different symmetries are observed. Looking forward to the follow-up work!

I am not completely convinced that the STT from the PML layer is completely washed out due to the 1st harmonic AC-detection. If present, it will induce torques which will modify the total response from the SOT's and other sources(?). But I agree with the authors that this is a delicate statement and for now the added paragraph discussing a possible STT from the PML makes this clear for the readers and provides a new avenue for investigation.

Overall, I now support publication as is.

Reviewer 1

The results will certainly initiate studies by other groups. While the authors used an optical technique to extract information on the magnetization of the Py layer, covered by a Pt layer, this is not useful for technological applications. Can the authors comment on how the stack can be implemented to readout the signal by electrical means.

The goal of this research is to demonstrate the spin-orbit effects with spin rotation symmetry, where MOKE is used for detecting the magnetization reorientation. In practical applications where we want to generate perpendicular spin to switch a perpendicular magnetization, we suggest use a different structure as discussed in the end of the paper, where the perpendicularly polarized spin current should be generated by an in-plane magnetization and the magnetization to be switched should be perpendicular. In order to read out the perpendicular magnetization orientation, one can build a magnetic tunnel junction on top of the perpendicular magnetization with a reference layer also having a perpendicular magnetization. The information embedded in the magnetization can be read out by measuring the tunneling magnetoresistance.

Reference: Liu, L. et al. Spin-Torque Switching with the Giant Spin Hall Effect of Tantalum. *Science* 336, 555-558 (2012).